# Study on Flavonoids and Bioactivity Features of Pericarp of *Citrus reticulata* “Chachi” at Different Harvest Periods

**DOI:** 10.3390/plants11233390

**Published:** 2022-12-05

**Authors:** Shejian Liang, Jiongbin Zhang, Yufang Liu, Zhijia Wen, Xinxin Liu, Fengliang Dang, Tianxiao Xie, Jingxin Wang, Zhanqian Wang, Hong Wu

**Affiliations:** 1Guangdong Technology Research Center for Traditional Chinese Veterinary Medicine and Natural Medicine, College of Life Science, South China Agricultural University, Guangzhou 510642, China; 2Laboratory for Lingnan Modern Agriculture, Guangzhou 510642, China

**Keywords:** Pericarpium Citri Reticulatae Chachiensis, flavonoid, metabolomics, antioxidation, SARS-CoV-2

## Abstract

Dry mature pericarp of *Citrus reticulata* “Chachi” (PCR), Pericarpium Citri Reticulatae Chachiensis, is a traditional Chinese medicine that displays characteristics of different usage at different harvest times in clinical use. The corresponding changes in the bioactive components in PCR from different harvest times remain unclear. Therefore, in this study, broadly targeted metabolomics technology was used to compare the differences in bioactive components among pericarps of PCR, which are the raw material of PCR at different growth stages. In the results, 210 kinds of flavonoid metabolites were detected. The content of hesperidin in red PCR harvested in December was higher than that in Citri Reticulatae Pericarpium Viride (CRPV) and reddish PCR harvested from July to November. Furthermore, the content of nobiletin, tangeretin, and 3,3′,4′,5,6,7,8-heptamethoxyflavone in CRPV from July to September was higher than that in the PCR harvested at other times. In addition, the result of cluster analysis and PCA showed that CRPV harvested from July to September had an obvious grouping pattern with the reddish PCR and the red PCR harvested from October to December. Differential metabolites in six comparison groups (A1 vs. A6, A1 vs. A2, A2 vs. A3, A3 vs. A4, A4 vs. A5, A5 vs. A6) were 67, 48, 14, 51, 42, and 40, respectively. The common differential metabolite of four comparison groups was 3′,4′,7-trihydroxyflavone (A1 vs. A2, A2 vs. A3, A3 vs. A4, A4 vs. A5). All the flavonoid differential metabolites screened were enriched in 16 metabolic pathways. Moreover, the results of the evaluation of the total antioxidant capacity indicated that CRPV in August was a suitable raw material for the production of antioxidants. Through molecular docking, the content of potential anti-SARS-CoV-2 components in the PCR in October was higher than that in the PCR in other periods. These results further proved that PCR at different harvest times was endowed with different efficacy and usage due to the difference in the accumulation of bioactive components.

## 1. Introduction

The dry mature pericarp of *Citrus reticulata* “Chachi” (PCR), Pericarpium Citri Reticulatae Chachiensis, is one of the top ten most popular traditional authentic medicinal materials in Guangdong Province and is mainly produced in Xinhui, Guangdong Province. According to traditional Chinese medicine, PCR is beneficial for the spleen and lungs, has a pungent taste, and is mild in nature and non-toxic. In addition, PCR has the effects of regulating Qi-flow to strengthen the spleen, harmonize the stomach, prevent vomiting, and dries dampness to reduce phlegm. It is mainly used for treating heaviness in the chest, bloating in the abdomen, coughing with excessive phlegm, reduced appetite, vomiting, and diarrhea, and has the functional characteristics of homology of medicine and food [1]. PCR is rich in essential oils and flavonoids. Flavonoids possess as many as 219 kinds of metabolites and mainly include flavonoid glycosides and polymethoxyflavones [2,3,4]. In recent years, some studies have indicated that flavonoids in PCR demonstrate not only anti-inflammatory, antioxidant, and lipid-lowering pharmacological effects but also pharmacological activities such as intestinal immune regulation, inhibition of adipocyte differentiation, and reduction in cancer cell proliferation [5,6,7,8,9,10]. Because of its positive health effects, PCR has great development potential in drugs, condiments, medicated meals, and health food. Currently, market demand for PCR is high, and the planting regions for *Citrus reticulata* “Chachi” are also expanding. The market scale of PCR has reached RMB 10 billion (USD 1.4 billion) [11].

To date, the novel coronavirus discovered at the end of 2019 has caused more than 539 million infections and more than 6.32 million deaths worldwide. However, there is still no specific drug or treatment for the disease caused by the novel coronavirus [12,13,14]. Adopting multiple ways to prevent the spread of the novel coronavirus is an important means to effectively control the prevalence of COVID-19 [15,16]. There are some studies that have shown that flavonoids such as neohesperidin, hesperidin, baicalin, kaempferol 3-rutinoside, and rutin from different sources with antiviral, antibacterial, and anti-inflammatory activities could effectively interact with these targets of novel coronavirus, of which several important flavonoid glycosides and polymethoxyflavones in PCR have binding affinity with SARS-CoV-2 target proteins [4,17].

According to the needs of medicinal use, PCR was divided into three kinds: Citri Reticulatae Pericarpium Viride (CRPV) harvested from May to August, reddish PCR harvested from October to November, and red PCR harvested from December to January [18]. CRPV is physiologically immature with a cyan appearance, a thin hard skin and a pungent and fragrant taste. Reddish PCR is generally harvested from 8 October to 23 November. Its physiology is not fully mature and its appearance is brownish yellow with thick and hard skin. It is pungent and slightly sweet in taste. According to traditional Chinese medicine, its medicinal effect is relatively mild. Red PCR is generally harvested from 22 November to 6 January. It is fully mature physiologically and appears brownish red in appearance. It has the characteristics of soft texture, thick skin, and a spicy, fragrant, and sweet taste [18]. The characteristics of the PCR are recorded in the *Compendium of Materia Medica*: “The red PCR floats and rises, enters the spleen and lung Qi; CRPV sinks and falls, enters the liver and gallbladder Qi”. This means that as medicinal materials with the same source and medicinal parts, PCRs have different functions due to different harvesting dates [19]. According to traditional Chinese medicine, CRPV has a strong medicinal effect. It is mainly used to soothe the liver, break the Qi, and help digestion, and for hypochondriac pain [18,20,21]. The effect of reddish PCR is similar to that of CRPV, mainly for helping digestion, soothing the liver, and breaking Qi, and the nutrients of reddish PCR are beneficial to the human body [18]. Red PCR is mainly used for regulating Qi, strengthening the spleen, drying dampness, and resolving phlegm [22]. Recently, Zeng et al. [23] found that, in the study on the correlation between the dynamic changes in flavonoids of PCR at different harvest times and its anti-lipase activity, the total content of six kinds of flavonoids (nobiletin, hesperidin, naringin, tangeretin, didymin, 3,3′,4′,5,6,7,8-heptamethoxyflavone) in PCR decreased from 5.663 ± 0.792% to 3.593 ± 0.670% with the delay of harvest time, which was from August to December. On the contrary, Wang et al. [24] reported that with the delay of the harvest period, which is from October to December, the contents of hesperidin and tangerine in *Citrus reticulata* “Chachi” increased, while the contents of nobiletin decreased. The fruits of *Citrus reticulata* “Chachi” at different harvest times have different health-care or medicinal values, and it is related to its bioactive components contained in the fruits at different harvest times. Therefore, a systematic and comprehensive study on the composition and law governing changes in flavonoids in the key period of fruit of *Citrus reticulata* “Chachi” development is significant for scientific harvesting of high-quality fruits. However, in most previous studies, the research objectives were mostly focused on several commonly reported flavonoid compounds and most of the research methods on the rules governing changes and metabolic pathways of flavonoid metabolites in the PCR at different developmental stages were traditional targeted metabolite-detection techniques.

Extensive targeted metabolomics analysis is a new method that combines the advantages of non-targeted metabolomics and targeted metabolomics. It can identify metabolites in samples by using the UPLC-QTrap-MS/MS detection platform, the self-built secondary database, and the multiple reaction monitoring (MRM) mode. It has the advantages of high throughput, high sensitivity, and accurate qualitative analysis [25,26,27]. The results of studies carried out using metabolome technology revealed that the content of flavonoids in reticulate PCR would change at different ages and different harvest periods, which provided a basis for further research on the accumulation dynamics and quality evaluation of flavonoids in PCR [4,28,29]. In order to comprehensively reveal the rules governing the change and bioactivity characteristics of flavonoids in reticulate PCR at different growth stages, this study used widely targeted metabolomics technology to carry out comparative metabolomics research on flavonoids in PCR that was harvested from July to December and detected the types, relative contents, and metabolic pathways of flavonoids in CRPV, reddish PCR, and red PCR. At the same time, this study carried out in vitro antioxidant activity detection and antiviral potential evaluation of PCR in different harvest periods. The research results have important theoretical and practical significance for the evaluation of PCR, in-depth research of its medicinal components, and scientific harvesting. It also provides a scientific basis for the phenomenon of PCR that has different efficacy and usage, which is listed in the *Compendium of Materia Medica*.

## 2. Results

### 2.1. Determination of Total Flavone Content

Figure 1 shows the standard curve results of the rutin standard. Figure 2 shows the results of the total flavonoids of tangerine peel at six different harvesting periods according to the standard curve of the rutin standard. It shows the content of total flavonoids of tangerine peel at six different harvesting periods, which was within the range of 0.45–0.85%.

The content of total flavonoids in six groups of PCRs at different harvest times was determined by colorimetry. The results showed that the content of total flavonoids increased first and then decreased with the delay of harvest time. The total flavone content of PCR harvested in August was the highest and that in December was the lowest. This meant that the total flavone content of CRPV collected in August was at its highest and then it decreased continuously in the process of transforming to reddish PCR and red PCR.

### 2.2. Metabolic Characteristics of Flavonoids

#### 2.2.1. Qualitative and Quantitative Analysis of Flavonoid Metabolites

In order to evaluate the effect of harvest time on flavonoids in PCR, an extensive targeted metabolomic analysis method was used to analyze the flavonoid metabolites in six groups of PCR samples at different harvest times. Figure 3A,B is diagrams of the total ion current (TIC) of QC samples detected by mass spectrometry, which are spectrum diagrams obtained by continuously plotting the intensities of all ions in the mass spectrum at each time point. Based on the local metabolism database, the flavonoid metabolites in the samples were analyzed qualitatively and quantitatively by mass spectrometry. Figure 3C,D shows multi-peak chromatograms of metabolites of QC samples using multiple-reaction monitoring, where the abscissa is the retention time of metabolite detection, and the ordinate is the ion current intensity of ion detection. Figure 3C,D shows the flavonoid metabolites that can be detected in the sample, where each mass spectrum peak with different colors represents one substance detected. In this study, 210 flavonoid metabolites were detected, including 90 flavones, 44 flavonols, 30 flavone glycoside, 19 dihydroflavones, 9 isoflavones, 6 dihydroflavonols, 6 flavanols, 2 chalcones, 1 sinensetin, 1 anthocyanin, and 2 other flavonoid compounds. Appendix A presents information about the detected metabolites, including precursor ions, integral values, and corresponding metabolite names.

The repeatability of metabolite extraction and detection could be judged by overlapping display analysis of TIC maps of mass spectrometry detection of different QC samples, which meant technical repetition. Figure 3E,F is superposition diagrams of the TIC diagram detected by mass spectrometry of QC samples. The results showed that the total ion current curves of metabolite detection had high overlap, which meant that the retention time and peak intensity were consistent. It indicated that the signal stability was good when mass spectrometry detected the same sample at different times. High stability of the instrument provided an important guarantee of the repeatability and reliability of the data.

#### 2.2.2. Multivariate Analysis of Metabolites

A cluster heat map was constructed by using the ionic strength data of flavonoid metabolites. The range method was used to normalize the content data of the 210 kinds of flavonoid metabolites detected. R software (Version number: 3.6.1, www.r-project.org/ (accessed on 5 August 2019)) was used to carry out cluster analysis of the accumulation pattern of metabolites among different samples, and the R program script was used to draw the cluster heat map. The cluster analysis results are shown in Figure 4. The results showed that the flavonoid metabolites of PCR gradually changed with the passage of harvest time.

In this experiment, principal component analysis (PCA) was performed on all samples. The PCA scores of each group of samples and QC samples are shown in Figure 5, where mix is the QC sample. The separation trend of flavonoid metabolome among the groups in this experiment was obvious, indicating that there were certain differences between the samples. PCA results showed that there were obvious differences between the flavonoid metabolites of PCR collected from July to September and from October to December. It was consistent with the above clustering analysis results. The results of five QC samples (mix01–05) basically overlapped, which meant that the QC samples had repeatability. 

To summarize, the clustering results of the samples showed that six groups of different harvest times clustered into two classes. A1, A2, and A3 clustered into one class and A4, A5, and A6 clustered into the other, which meant that the CRPV collected from July to September could be distinguished from the reddish PCR and the red PCR collected from October to December by the type and characteristics of flavonoid metabolites. The PCA score results showed that the separation trend of flavonoid metabolome between groups was obvious, indicating that there were certain differences between samples. Moreover, the PCA scores showed that there were obvious differences between flavonoid metabolites of PCR collected from July to September and from October to December, which was consistent with the cluster analysis results. In addition, five QC samples (mix01–05) basically overlapped, showing good QC sample repeatability. These results showed that the data processing results of this experiment were reliable.

### 2.3. Identification, Screening, and Analysis of Flavonoid Metabolites

By means of clustering analysis and PCA of six groups of PCR samples, it was found that the flavonoid metabolites significantly varied over the harvest period. The difference in the expression level of differential metabolites in each comparison group and the statistical significance of the difference can be quickly viewed through the volcano map. Comparing the ion strength of flavonoids in PCR samples with different harvest times, differential metabolites were screened out. The results showed that there were 67, 48, 14, 51, 42, and 40 differential metabolites screened by A1 vs. A6, A1 vs. A2, A2 vs. A3, A3 vs. A4, A4 vs. A5, and A5 vs. A6, respectively (Figure 6). 

The fold change diagram was employed to reflect the flavonoid metabolites that change greatly in each compared group after the difference fold log_2_ treatment. Figure 7 presents a fold change diagram of flavonoid metabolites that change greatly in each compared group. The results showed that, from the perspective of the whole harvesting period (A1 vs. A6), the types of up-regulated differential metabolites were mainly flavone and dihydroflavonoid glycosides, of which Chrysoeriol-7-O-[β-D-glucuronopyranosyl-(1→2)-O-β-D-glucuronopyranoside] had the highest up-regulation multiple, which was 11.84. The down-regulated differential metabolites were mainly flavone, flavanols, and flavonols, of which 7,3′,4′-trihydroxyflavone had the largest down-regulation multiple, which was 4.15, and other differential metabolites had a down-regulation multiple of less than 4. 

To identify the differential metabolites from the flavonoid metabolites among each comparison group, a Venn diagram was employed (Figure 8). Figure 8 presents a Wayne diagram of the relationship between the differential metabolites of each comparison group. It was found that in the four comparison groups of A1 vs. A2, A2 vs. A3, A3 vs. A4, and A4 vs. A5, there was a common differential metabolite, 7,3′,4′-trihydroxyflavone. The expression of this flavonoid was down-regulated in PCR that was harvested from July to October and up-regulated in PCR that was harvested from October to November. In the three comparison groups of A3 vs. A4, A4 vs. A5, and A5 vs. A6, there were five common differential metabolites, including two flavone glycosides (orientin, isoorientin), two flavones (luteolin, chrysoeriol), and one flavonol (isorhamnetin). The expression of these five flavones in PCR collected from September to November showed a downward trend, and they all showed an upward trend in PCR collected from November to December.

### 2.4. Analysis of Main Flavonoid Monomers

Flavonoids are the main active components in PCR, including flavonoid glycosides and polymethoxyflavones. In total, 210 kinds of flavonoid metabolites were detected in the samples. Figure 9 shows the differences in the contents of hesperidin, nobiletin, tangeretin, and 3,3′,4′,5,6,7,8-heptamethoxyflavone substances in the PCR at different harvest times. As shown in Figure 9, with the extension of the harvest period from July to December, the content of hesperidin decreased first and then increased, reaching the maximum in December, which meant that the content of hesperidin in the red PCR was the highest. The content of nobiletin increased first and then decreased with the extension of the harvest period from July to December, reaching the maximum and minimum in September and December, respectively. This meant that the content of nobiletin in CRPV harvested in September was the highest. The contents of tangeretin and 3,3′,4′,5,6,7,8-heptamethoxyflavone decreased with the delay of harvest time, and also reached the highest level in the CRPV.

### 2.5. Evaluation of Total Antioxidant Capacity

To investigate the impact of the change in flavonoid content in PCR during harvest times on bioactivity, FRAP and ABTS methods were employed to determine the total antioxidant activity of the extract solutions of PCR in six groups with different harvest times (Table 1). With the extension of harvest time, the reducing ability of FRAP of PCR increased first and then decreased, which meant that the antioxidant activity of PCR increased first and then decreased and the antioxidant activity of PCR collected in August was the highest, while that in November was the lowest. With the extension of harvest time, the free radical scavenging activity of ABTS of PCR increased first and then decreased, which also meant the antioxidant activity of PCR increased first and then decreased and the antioxidant activity of PCR collected in August was the highest, while that in December was the lowest.

### 2.6. Molecular Docking Analysis

In order to further understand the role of PCR in the prevention and treatment of COVID-19 from the perspective of modern medicine, molecular docking was used to evaluate the binding energy of flavonoids and positive control drugs separately for protein structures of spike, 3CLpro, PLpro, and RdRp in this study according to the results of a previous molecular docking technology study [4]. The docking results of the positive control drug molecules are shown in Table 2. Lopinavir had the lowest binding energy at 3CLpro, RdRp, and spike proteins, with values of −6.20 kcal/mol, −10.03 kcal/mol, and −11.40 kcal/mol, respectively. Ribavirin had the lowest binding energy for PLpro, with a value of −7.27 kcal/mol. 

Table 3 lists 26 components from PCR that had lower binding energy for 3CLpro than lopinavir. In them, isoxaferoside had the lowest binding energy of –9.42 kcal/mol. In addition, some flavonoids abundant in PCR had lower binding energy than lopinavir, e.g., hesperidin −6.43 kcal/mol, naringin −7.83 kcal/mol, narirutin −8.93 kcal/mol, neohesperidin −8.33 kcal/mol, nobiletin −6.80 kcal/mol, and tangeretin −6.83 kcal/mol. 

Table 4 lists 11 components from PCR that had lower binding energy for RdRp than lopinavir. Linarin had the lowest value of −11.97 kcal/mol. Moreover, some flavonoids abundant in PCR had lower binding energy than lopinavir, e.g., naringin −10.83 kcal/mol, narirutin −11.63 kcal/mol, and neohesperidin −10.33 kcal/mol. 

Table 5 lists 11 components from PCR that had lower binding energy for PLpro than Ribavirin. Neohesperidin had the lowest value of −8.00 kcal/mol. 

Table 6 lists five components of PCR which had lower binding energy for spike than lopinavir. Isoxaphoroside had the lowest value of −13.33 kcal/mol. In addition, lonicerin, naringin, isorhoifolin, and hesperidin had values of −11.90 kcal/mol, −11.83 kcal/mol, −11.70 kcal/mol, and −11.70 kcal/mol, respectively.

To compare potential antivirus activities of PCR with different harvest times against SARS-CoV-2, the total content of flavonoids with lower binding energy than the positive control drug was for the sum of the separate targeting protein and the separate group of harvest time. Figure 10A shows the total content of 26 flavonoids with lower binding energy than the positive control drug for 3CLpro with harvest time. It was found that the content of group A4 was the highest. Figure 10B–D show the results for binding target proteins RdRp, PLpro, and spike, respectively. In summary, the screened flavonoids with lower binding energy to the four target proteins than the positive control might have better anti-SARS-CoV-2 ability. The higher the content of these flavonoids in PCR was, the stronger the anti-SARS-CoV-2 ability was. Generally, the total content of flavonoids with lower binding energy than the positive control drug was highest in group A4, which indicated that PCR collected in October had strong anti-SARS-CoV-2 ability.

### 2.7. KEGG Annotation and Enrichment Analysis of Flavonoid Differential Metabolites

In order to explore the metabolism of flavonoids in PCR at different harvest times, the KEGG database was used to annotate the detected differential metabolites and analyze the enrichment of pathways. The annotation results of the flavonoid differential metabolite KEGG in each comparison group were classified according to the pathway types in KEGG [30]. The results of KEGG classification and enrichment analysis are shown in Figure 11. Among the 67, 48, 14, 51, 42, and 40 flavonoid differential metabolites screened in the six comparison groups (A1 vs. A6, A1 vs. A2, A2 vs. A3, A3 vs. A4, A4 vs. A5, A5 vs. A6), 8, 12, 2, 14, 5, and 6 were annotated into the corresponding metabolic pathways, that is, 8 pathways in A1 vs. A6, 11 pathways in A1 vs. A2, 1 pathway in A2 vs. A3, 6 pathways in A3 vs. A4, 14 pathways in A4 vs. A5, and 13 pathways in A5 vs. A6. All the screened flavonoid differential metabolites were enriched in 16 metabolic pathways. The analysis showed that some flavonoid differential metabolites could participate in multiple metabolic pathways and there were multiple metabolic pathways consistent with the comparison groups.

Figure 12 shows the results of KEGG pathway enrichment, which were gathered according to the annotation results of a flavonoid differential metabolite in each comparative group of KEGG. The Rich factor is the ratio of the number of differentially expressed metabolites in the corresponding pathway to the total number of metabolites detected and annotated by the pathway. The larger the value is, the greater the enrichment degree is. The closer the *p* value is to 0, the more significant the enrichment is. The flavonoid differential metabolites screened from each comparison group were significantly enriched in many pathways, including flavone and flavonol biosynthesis pathways, phenylpropane biosynthesis pathways, flavonoid biosynthesis pathways, phenylalanine biosynthesis pathways, tyrosine and tryptophan biosynthesis pathways, biosynthesis of antibiotics pathways, degradation of aromatic compounds pathways, aminobenzoate degradation pathways, polycyclic aromatic hydrocarbon degradation pathways, and benzoate degradation pathways.

## 3. Discussion

### 3.1. Harvesting Periods Affected the Accumulation of Bioactive Components in PCR

PCR is divided into three kinds according to the harvest time in production: CRPV harvested from May to August, reddish PCR harvested from October to November, and red PCR harvested from December to January [18]. As mentioned above, the characteristics of PCR are recorded in the *Compendium of Materia Medica*: “PCR floats and rises, enters the spleen and lung Qi; CRPV sinks and falls, enters the liver and gallbladder Qi”. This means that PCR medicinal materials with the same source and medicinal parts have different functions due to different harvesting periods [19]. According to traditional Chinese medicine, CRPV has a strong medicinal effect and is mainly used to soothe the liver, break the Qi, and help digestion, and for hypochondriac pain [18,20,21]. The effect of reddish PCR is similar to that of CRPV, mainly for eliminating accumulation and stagnation, soothing liver, and breaking Qi and reddish PCR is very nutritious [18]. Red PCR is mainly used for regulating Qi, strengthening the spleen, drying dampness, and resolving phlegm [22]. It has been reported that the differences in the content and proportion of flavonoids in the PCR at different harvest times may lead to differences in drug properties and efficacy, which indicates that PCR has the same origin but different usage [31]. In order to comprehensively and deeply explore the dynamic changes in the content and composition of flavonoids in PCR at different harvest periods, extensive targeting technology was used to determine the flavonoids of PCR in different harvest periods, in which hesperidin, nobiletin, tangeretin, 3,3′,4′,5,6,7,8-heptamethoxyflavone were focused on and analyzed. It was found that the content of hesperidin decreased first and then increased with the extension of the harvest time, reaching the maximum in December. The content of nobiletin increased first and then decreased, reaching the maximum and minimum in September and December, respectively. However, the contents of tangeretin and 3,3′,4′,5,6,7,8-heptamethoxyflavone decreased with the extension of the harvest time. In the *Chinese Pharmacopoeia* (2020 edition), hesperidin, nobiletin, and tangeretin were specified as the index ingredients of PCR [1]. By evaluating the qualitative and quantitative analysis results of flavonoid metabolites in PCR at different harvest times, it was found that the content of hesperidin in the red PCR was higher than that in CRPV and reddish PCR. Moreover, the contents of three kinds of polymethoxyflavones (nobiletin, tangeretin, 3,3′,4′,5,6,7,8-heptamethoxyflavone), which had strong biological activities that were commonly reported in CRPV, were generally higher than those in reddish PCR and red PCR. At the same time, through cluster analysis and principal component analysis, it was found that the flavonoid metabolites of PCR collected from July to September and from October to December showed an obvious grouping pattern, which meant there were clear differences in the composition and content of flavonoid compounds between the CRPV collected from July to September and the reddish PCR and red PCR collected from October to December. The biosynthesis of secondary metabolites largely depended on cell type, developmental stage, and environmental cues [32]. Plant harvesting season and growth stage all affected the content of SMS in medicinal plants [33,34,35]. According to all the study results and a comprehensive analysis of the literature, it can be inferred that the flavonoid compounds synthesized in PCR have changed regularly in composition and content at different developmental stages. It is these differences in material basis that may form the phenomenon of PCR coming from the same raw materials at different growth stages but having different usage and different pharmacodynamics and drug properties.

### 3.2. The Difference in Bioactive Components of PCR in Different Harvest Periods Affected Its Antioxidant and Antiviral Effects

The bioactive components contained in medicinal plants are the material basis of their clinical therapeutic effects, and changes in their composition and content will directly affect their clinical efficacy. In this study, the antioxidant activities of PCR collected at different harvest times were compared in order to explore the characteristics of change in pharmacological activities of PCR collected at different harvest times. With the extension of harvest time, the reducing ability in FRAP and the free radical scavenging activity in ABTS of PCR increased first and then decreased, reaching the maximum in August. The results of FRAP showed that the antioxidant activity of PCR that was harvested in November was the lowest. The results of ABTS showed that the antioxidant activity of PCR that was harvested in December was the lowest. Citrus flavonoids have a variety of health-promoting functions, such as anti-inflammatory, anticancer, bacteriostatic, and antilipid effects, which might be based on their strong antioxidant activity [36]. According to the literature and these research results, it can be inferred that the CRPV in August is suitable raw material for producing antioxidants because the total flavone content of PCR is the highest at this time.

In the literature, there is research that conducted virtual molecular docking between the important flavonoid components in PCR of different ages and novel coronavirus spike protein, novel coronavirus RNA polymerase, and important proteases such as 3CLpro and PLpro. The results of virtual molecular docking in this research suggested that the binding affinity of many flavonoids in PCR, especially in fresh PCR, to four important functional proteins of SARS-CoV-2 was significantly better than that of positive control drugs, as in the results of [4]. The potential molecular mechanism of PCR interfering with COVID-19 might be due to the combination of PMFs with 3CL^pro^ (3CL hydrolase) against SARS-CoV-2, which affects the virus replication process [37]. According to these studies, the anti-novel coronavirus effects of flavonoids in fresh PCR collected at different harvest times were further analyzed and compared by molecular docking in this study. The results of molecular docking showed that the content of potential components against SARS-CoV-2 in PCR harvested in October was higher than that in PCR harvested in other periods. Nobiletin, 3,3′,4′,5,6,7,8-heptamethoxyflavone, and tangeretin of PMFs had certain inhibitory effects on 3CL^pro^. Because of its strong activity, 3,3′,4′,5,6,7,8-heptamethoxyflavone could be used as an excellent potential lead compound against SARS-CoV-2 [38]. Therefore, it is more appropriate to use reddish PCR collected in October as a herbal medicine for medical use to prevent COVID-19 because of its high content of flavonoids, with potential anti-SARS-CoV-2 activity. However, this study only determined the anti-SARS-CoV-2 effect of flavonoids of PCR by comparing the binding affinity, which was based on the results of molecular docking. It is necessary to carry out anti-SARS-CoV-2 experiments in vivo and in vitro separately.

### 3.3. Differences in Flavonoid Metabolites between PCRs Harvested in Different Growth Stages

The synthesis of flavonoids in citrus can be divided into three stages: synthesis of precursor substances (coumaroyl-CoA); synthesis of flavonoids such as flavone, flavanone, flavonols, and anthocyanins; and glycosylation to form various glycosides. Flavonoid compounds such as flavone, flavonols, isoflavones, flavanone, and anthocyanin have various physiological activities, including anti-inflammation, antioxidant, antiatherosclerotic, and antitumor. These flavonoid compounds are formed in plants, starting from phenylalanine produced in the shikimic acid pathway and progressing through several branching pathways of the phenylpropanoid metabolic pathway [39,40]. In this study, 67, 48, 14, 51, 42, and 40 kinds of differential metabolites were screened through OPLS-DA for six comparative groups (A1 vs. A6, A1 vs. A2, A2 vs. A3, A3 vs. A4, A4 vs. A5, A5 vs. A6), of which 8, 12, 2, 14, 5, and 6 kinds of differential metabolites, respectively, were annotated into corresponding metabolic pathways, mainly in flavone and flavonol biosynthesis, phenylpropane biosynthesis, flavonoid biosynthesis, and phenylalanine biosynthesis. The phenylpropane metabolic pathway is an important secondary metabolic pathway in organisms, and all phenylpropane compounds containing the phenylpropane skeleton are direct or indirect products of this pathway. The research on the regulation of flavonoid biosynthesis in citrus is currently still in its infancy [9,41]. Different metabolites of PCR collected at different harvest times are significantly enriched in these pathways, which is helpful to further investigate the metabolic pathways of the active substances of PCR, to understand their intermediate metabolites and regulatory mechanisms, and provide a theoretical basis of the use of PCR in traditional Chinese medicine and research in the field of food health care.

## 4. Materials and Methods

### 4.1. Plant Materials

The raw materials of *Citrus reticulata* “Chachi” were collected in the standardized planting demonstration base of PCR in Xinhui District, Jiangmen City, Guangdong Province from July to December 2017. Table 7 shows the detailed harvesting time and sample information. Information about fresh fruits is shown in Figure 13. The images (a1) to (a6) of Figure 13 represent the appearance of fruits of *Citrus reticulata* “Chachi” at different harvesting times. All the pericarp of *Citrus reticulata* “Chachi” that was harvested in different harvest periods was peeled and exposed to the sun for 48 h before being stored in a cool and dry place. Information about the samples of PCR is presented in Figure 14. PCRs were harvested and processed in each month in three biological replicates, each numbered A1-1-3, A2-1-3, A3-1-3, A4-1-3, A5-1-3, and A6-1-3. There were six comparison groups in this study: A1 vs. A6, A1 vs. A2, A2 vs. A3, A3 vs. A4, A4 vs. A5, A5 vs. A6. According to the production demand standard, A1, A2, and A3 harvested from July to September belonged to CRPV; A4 and A5 harvested from October to November belonged to reddish PCR; and A6 harvested from December belonged to red PCR.

### 4.2. Sample Preparation

#### 4.2.1. Sample Preparation and Extraction for Flavonoid Metabolomic Analysis

All samples were collected and lyophilized, and comminuted with a mixer (MM400, Retsch Technology, Haan, Germany) for 1.5 min at 30 Hz. Then, 100 mg powder of each sample was weighted and 1.0 mL 70% aqueous methanol was used for extraction. In order to assure complete extraction, the resulting mixtures were kept at 4 °C overnight and vortexed three times. Before LC-MS analysis, the extracts were centrifuged at 10,000× *g* for 10 min and then absorbed and filtrated (SCAA-104, 0.22 μm pore size).

#### 4.2.2. Quality Control (QC) Samples

The extracts of samples in all groups were blended with equal amounts in order to prepare QC samples. Five replicate QC samples, which were named mix01 to mix05, were prepared individually. We used the same method to determine all QC samples as well as the analytical samples. In order to evaluate the repeatability of the whole analytical process, one QC sample was determined after every three analytical samples in instrumental analysis.

### 4.3. Methods

#### 4.3.1. Determination of Total Flavonoid Content and Total Antioxidant Activity

PCR was powdered and sieved using 40 mesh sieves. Then, 1 g of PCR powder was accurately weighed into conical flasks, sealed with 50 mL 75% ethanol and extracted for 8 h. The mixture was treated with an ultrasonic cleaner at 45 °C and 100% power for 40 min. The solution was filtered into a 50 mL volumetric flask and the volume was fixed with 75% ethanol. Three replicate sample solutions were made separately from every PCR sample. The method of determination of total flavonoid content was made by modifying the method of Wang et al. [42] and Li et al. [43].

The detection kit from Beyotime Biotechnology Co. Ltd., Shanghai, China. was employed and used for the determination of total antioxidant activity, following the operational procedure of the FRAP and ABTS methods.

#### 4.3.2. HPLC Conditions

An LC-ESI-MS/MS system was used to analyze the samples. The whole analytical system conditions are described as follows. The HPLC column was purchased from Waters ACQUITY UPLC HSS T3 C18 (1.8 μm, 2.1 mm × 100 mm). The composition and proportion of the solvent system was water (0.04% acetic acid):acetonitrile (0.04% acetic acid). The gradient programming was set as follows: 0 min 100:0*v*/*v*, 11.0 min 5:95*v*/*v*, 12.0 min 5:95*v*/*v*, 12.1 min 95:5*v*/*v*, 15.0 min 95:5*v*/*v*. The temperature was 40 °C and the flow rate was set as 0.40 mL/min. The injection volume was 5 microliters. The effluent of the sample was alternatively connected to the ESI-triple quadrupole-linear ion trap (Q TRAP)-MS.

#### 4.3.3. ESI-Q TRAP-MS/MS

A triple quadrupole-linear ion trap mass spectrometer (Q TRAP) of API 6500 Q TRAP LC/MS/MS System, which was fit out using an ESI Turbo Ion-Spray interface, was used for LIT and triple quadrupole (QQQ) scans. It worked in a positive ion mode and was handled using Analyst software (Version number: 1.6.3, AB Sciex Pte. Ltd., Framingham, MA, USA). The operating parameters of the ESI source spray ion source are as follows. The ion spray voltage (IS) was 5500 V. The source temperature was set at 550 °C. The ion source gas I (GSI), gas II (GSII), and curtain gas (CUR) were set at 55, 60, and 25.0 psi, respectively. The collision gas (CAD) was rich. The ion source and the turbo spray were equipped. In QQQ and LIT modes, 10 and 100 μmol/L polypropylene glycol solutions were used for instrument tuning and mass calibration, respectively. QQQ scans were obtained as NMR experiments by setting the impact gas (nitrogen) at 5 psi. DP and CE for individual MRM transitions were finished through further DP and CE optimization. According to the metabolites that were eluted within this period, a specific set of MRM transitions were detected for each period.

### 4.4. Data Analysis

In this study, the metabolites were qualitatively analyzed by making use of the metabolite information in the MVDB V2.0 database and public database of Maiwei Biotechnology Co., Ltd. Existing mass spectrometry databases such as MassBank, KNAPSAcK, HMDB, and METLIN were used for primary and secondary mass spectrometry analysis. Triple quadrupole mass spectrometry multi-reaction monitoring mode (MRM) was used for quantitative analysis of metabolites. After obtaining the metabolite data of different samples, the software Analyst 1.6.1 was used for qualitative and quantitative mass spectrometry analysis, comprising baseline filtering, peak identification, integration, retention time correction, peak alignment, and mass spectrometry fragment attribution analysis. The data were normalized and annotated based on the obtained retention time, mass-to-charge ratio and peak intensity. A reliable mathematical model was established by multi-dimensional statistical analysis to analyze metabolites. PCA (principal component analysis) based on unsupervised pattern recognition was used to analyze the data of the detected metabolites, so as to preliminarily explore the overall metabolite difference between the samples in each group and the variability between the samples in the group. OPLS-DA (the orthogonal partial least squares discriminant analysis) of supervised pattern recognition was used to remove the irrelevant differences to screen the difference variables and establish the differences between the samples in each group. When the variable import in projection (VIP) was greater than 1, it was considered to be a difference variable. There were three prediction parameters of OPLS-DA model: R2X, R2Y, and Q2. The closer the three indexes were to 1, the more stable and reliable the model was. Multi-dimensional statistics VIP value (VIP > 1), single dimensional statistics (*p* < 0.05), and fold change were used to screen differential metabolites. After log2 transformation of fold change, the metabolites with VIP > 1, *p* < 0.05, log2 FC ≥ 2, or log2 FC ≤ 0.5 were selected as differential metabolites.

### 4.5. Kyoto Encyclopaedia of Genes and Genomes (KEGG) Annotation and Metabolism Pathway Analysis of Differential Flavonoid Metabolites

The Kyoto Encyclopedia of Genes and Genomes pathway database (http://www.kegg.jp/kegg/pathway.html (accessed on 28 August 2019)) was used to focus on metabolic reactions and connect possible metabolic pathways and corresponding regulatory proteins [44]. The key pathways with the highest differential correlation with metabolites could be further screened by enrichment analysis and topology analysis of pathways with differential metabolites [45].

## 5. Conclusions

In this study, the flavonoid metabolome of PCR at different harvest times was compared by using UPLC-QTrap-MS/MS technology. It was found that the flavonoid metabolome of CRPV harvested from July to September had a distinct grouping pattern compared with the flavonoid metabolome of reddish PCR and red PCR harvested from October to December. The CRPV harvested in August was found to be a suitable raw material for the production of antioxidants. It was more appropriate to choose reddish PCR harvested in October because of its high content of flavonoids with potential anti-SARS-CoV-2 activity when used as a herbal medicine to prevent COVID-19. The results of this study also proved that PCR at different harvest times had different bioactive effects and clinical uses due to the accumulation differences in bioactive components, and they provide a scientific basis for the phenomenon of PCR coming from the same raw material at different growth stages but having different usage, as described in the *Compendium of Materia Medica*.

## Figures and Tables

**Figure 1 plants-11-03390-f001:**
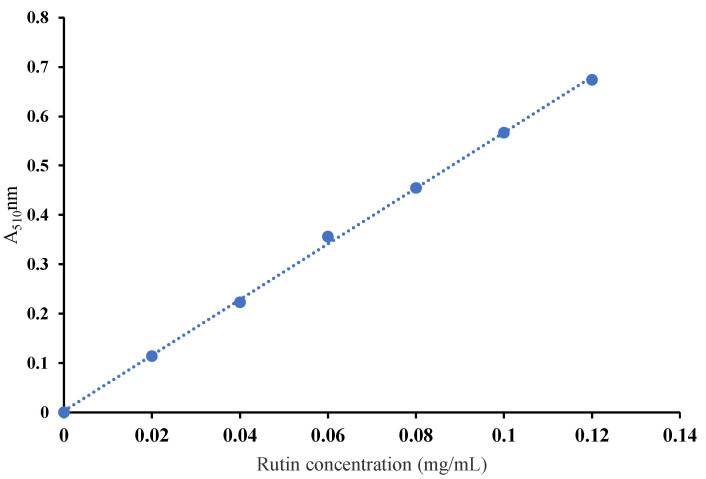
Rutin standard curve. The regression equation of the standard curve of the rutin standard is y = 5.6429x + 0.0027, R^2^ = 0.9992.

**Figure 2 plants-11-03390-f002:**
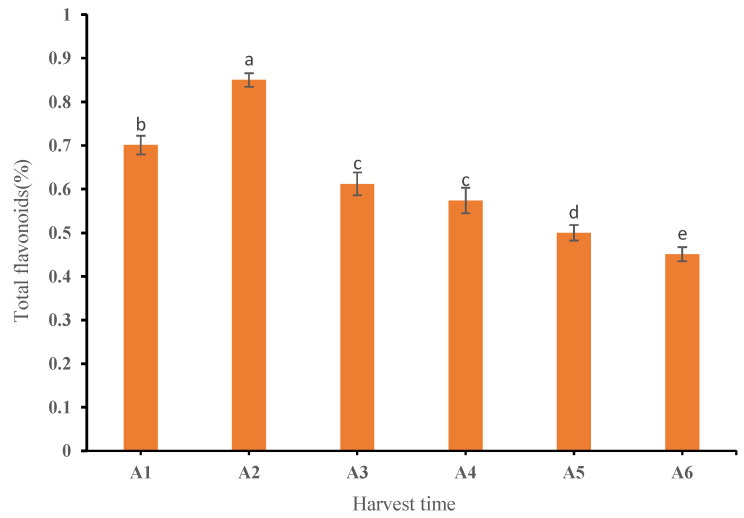
Change in total flavone content of dried PCR at different harvest periods. Different letters above each column indicate significant differences between the total flavone content of each month. The same letter above each column indicates no significant difference between the total flavone content of these months.

**Figure 3 plants-11-03390-f003:**
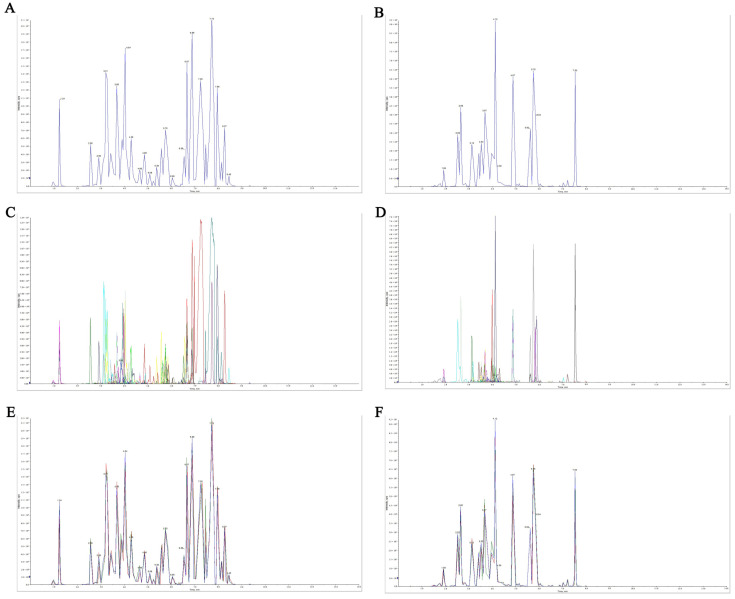
TIC diagram of QC sample mass spectrometry detection, multi-peak diagram of MRM metabolite detection and TIC superimposition diagram. (**A**,**B**) are the total positive ion current and total negative ion current diagrams detected by mass spectrometry of QC samples, respectively. (**C**,**D**) are MRM metabolite detection multi-peak diagrams of positive and negative ion flows of QC samples detected by mass spectrometry, respectively. (**E**,**F**) are superposition diagrams of the total positive ion current diagram and the total negative ion current diagram of QC sample mass spectrometry detection, respectively.

**Figure 4 plants-11-03390-f004:**
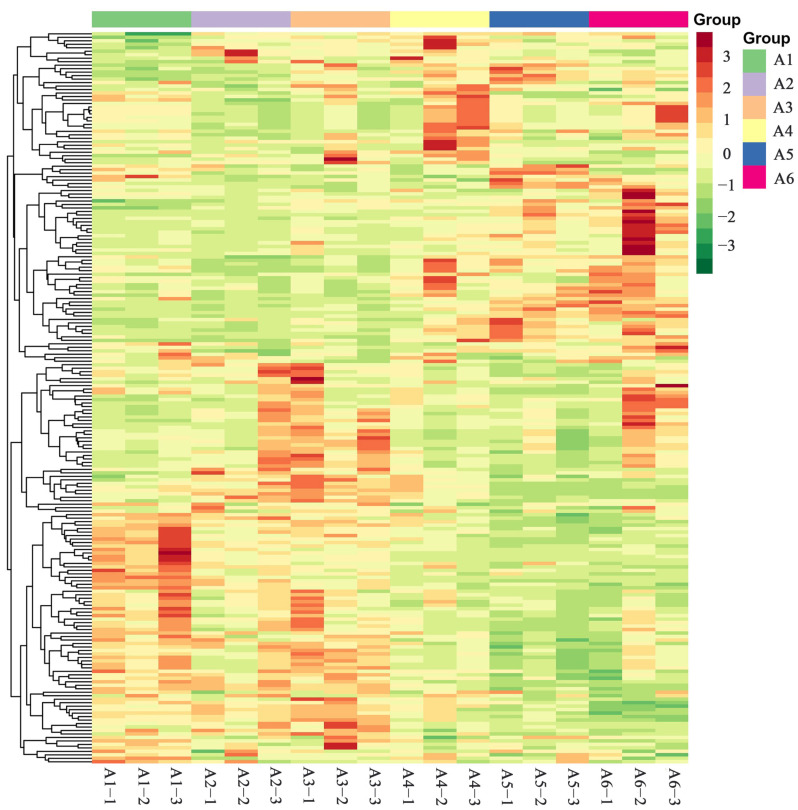
Heat map of cluster analysis of flavonoids in PCR at different harvest times. The abscissa represents the sample name and the ordinate represents the detected flavonoid metabolites. The order of color from orange to green indicates that the metabolite content gradually decreases.

**Figure 5 plants-11-03390-f005:**
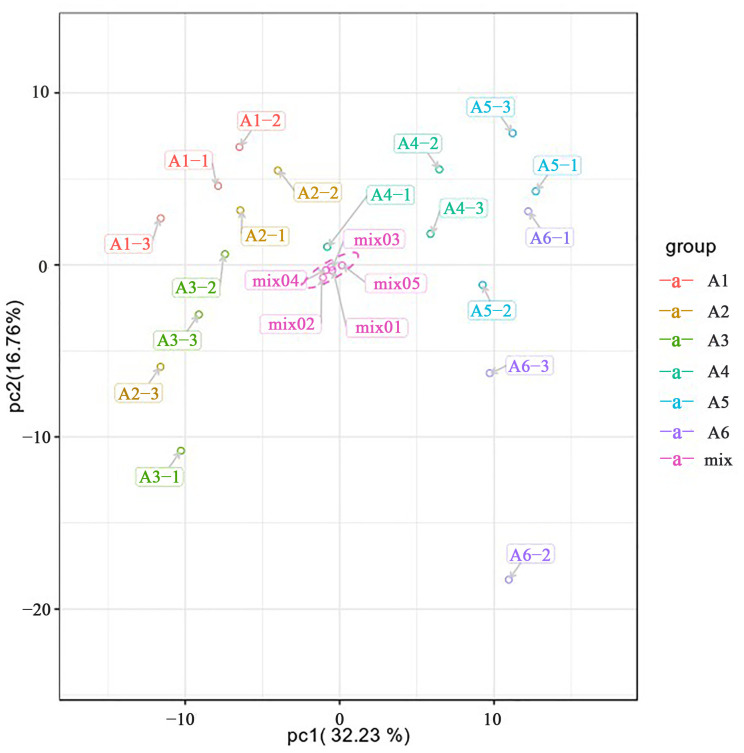
PCA scores of samples in each group and QC samples. The abscissa represents the first principal component and the ordinate represents the second principal component.

**Figure 6 plants-11-03390-f006:**
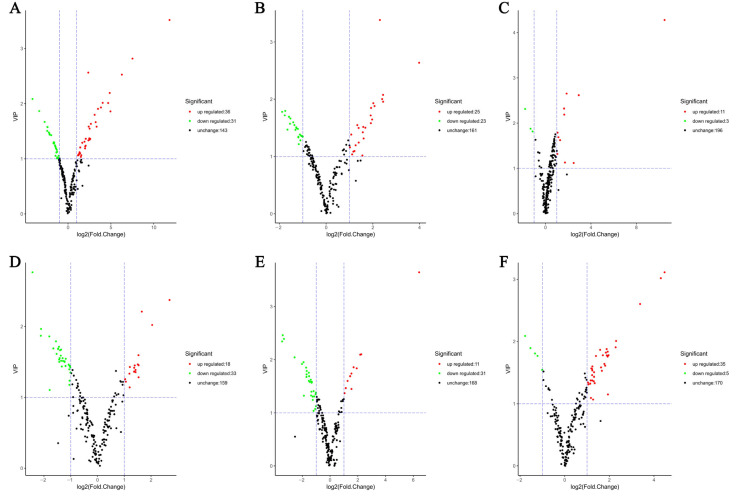
Volcano diagrams of differential metabolites. (**A**–**F**) Volcano plots of flavonoid differential metabolites of A1 vs. A6, A1 vs. A2, A2 vs. A3, A3 vs. A4, A4 vs. A5, and A5 vs. A6, respectively. Each point on the volcano map represents a metabolite, the abscissa represents the logarithm of the quantitative difference multiple of a metabolite in the two samples, and the ordinate represents the VIP value. The larger the absolute value of the abscissa is, the greater the fold difference of the expression between the two samples is. The larger the ordinate value is, the more significant the differential expression is, and the more reliable the differentially expressed metabolites are. The green dots in the figure represent down-regulated differentially expressed metabolites, while the red dots represent up-regulated differentially expressed metabolites, and the black dots represent detected but not significantly different metabolites.

**Figure 7 plants-11-03390-f007:**
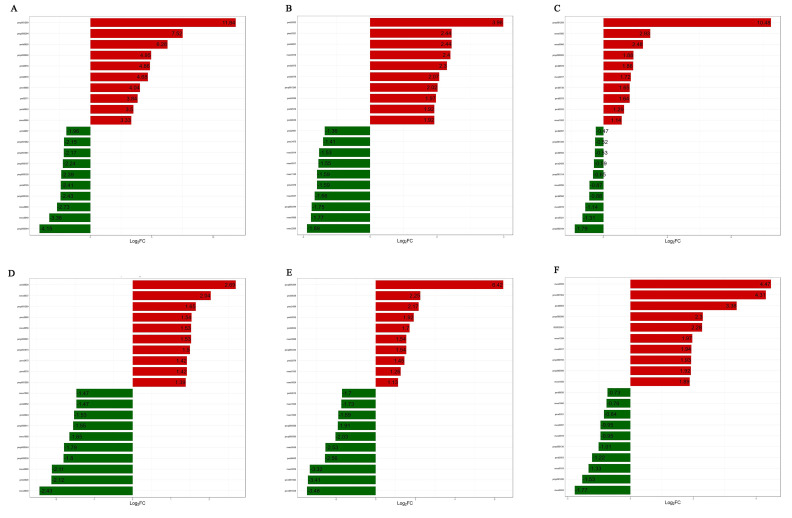
Fold change diagrams of flavonoid differential metabolites in each comparison group. (**A**–**F**) represent the differential fold diagrams of flavonoid differential metabolites of A1 vs. A6, A1 vs. A2, A2 vs. A3, A3 vs. A4, A4 vs. A5, and A5 vs. A6, respectively. The abscissa indicates that the different multiples are logarithm values based on 2 and the ordinate indicates the number of flavonoid differential metabolites.

**Figure 8 plants-11-03390-f008:**
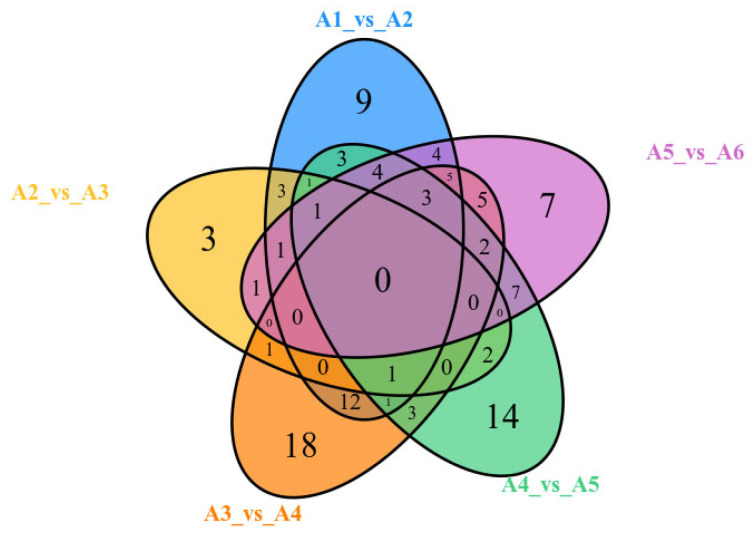
Wayne diagram of flavonoid differential metabolites in each comparison group.

**Figure 9 plants-11-03390-f009:**
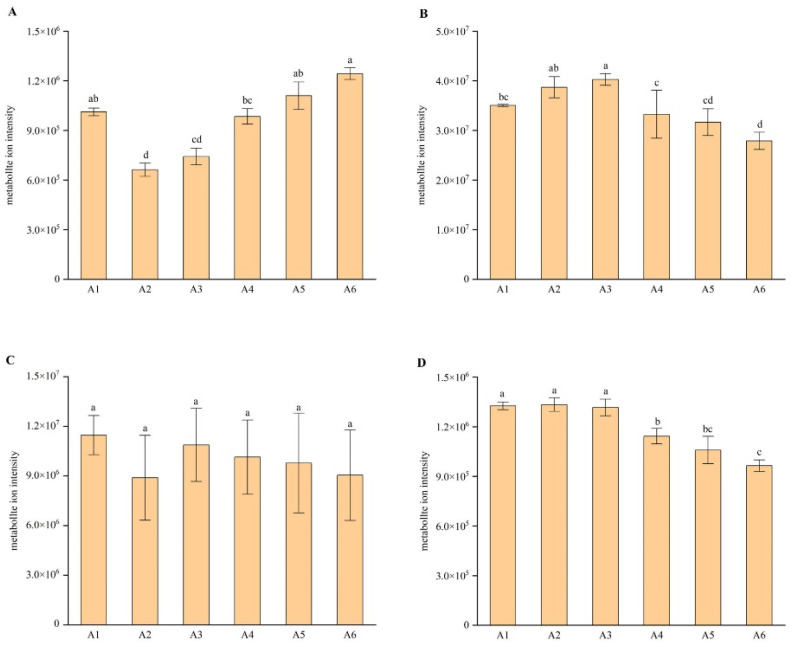
Integral correction results of quantitative analysis of four flavonoid compounds in different samples. The plots represent: (**A**) the difference in relative content of hesperidin in PCR at different harvest times; (**B**) the difference in relative content of nobiletin in PCR at different harvest times; (**C**) the difference in relative content of tangerine in PCR at different harvest times; (**D**) the relative contents of 3,3′,4′,5,6,7,8-heptamethoxyflavone in PCR at different harvest times. Different letters above each column indicate significant differences between the content of each flavonoid compounds of each month. The same letter above each column indicates no significant difference between the content of each flavonoid compounds of these months.

**Figure 10 plants-11-03390-f010:**
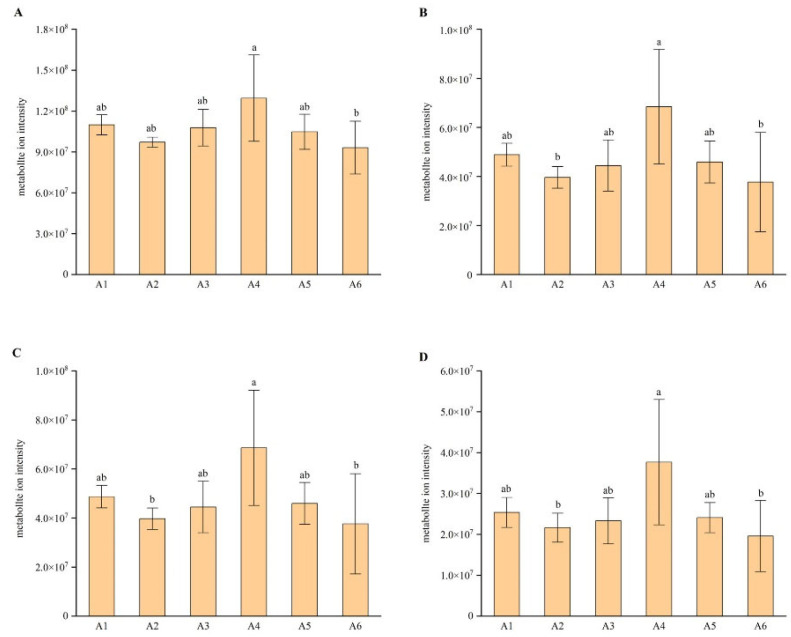
Change in total content of flavonoids with lower binding energy than positive control. The plots represent: (**A**) changes in the total contents of 26 flavonoids with lower binding energy to 3CLpro than to the positive control; (**B**) changes in the total contents of 12 flavonoids with lower binding energy to RdRp than to the positive control; (**C**) changes in the total contents of 11 flavonoids with lower binding energy to PLpro than to the positive control; (**D**) changes in the total contents of 5 flavonoids with lower binding energy to spike protein than to the positive control. Different letters above each column indicate significant differences between the total content of flavonoids which have lower binding energy than positive control of each month. The same letter above each column indicates no significant difference between the total content of flavonoids which have lower binding energy than positive control of these months.

**Figure 11 plants-11-03390-f011:**
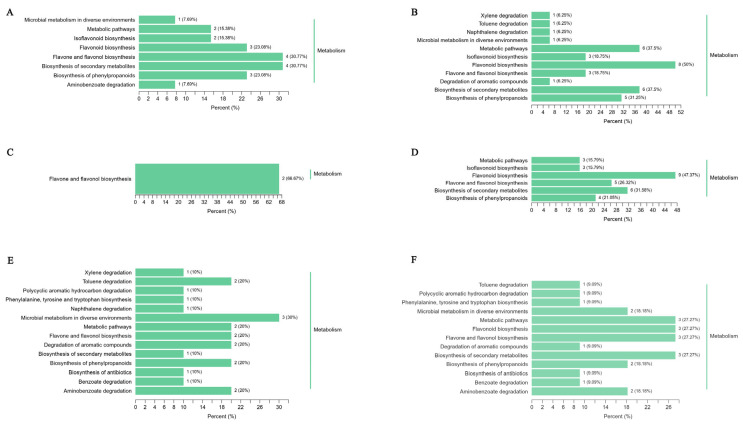
KEGG classification results of differential flavonoid metabolites in separate comparison pair groups. (**A**–**F**) Annotation results of KEGG of the flavonoid differential metabolites of A1 vs. A6, A1 vs. A2, A2 vs. A3, A3 vs. A4, A4 vs. A5, and A5 vs. A6, respectively. The abscissa is the number of metabolites annotated to this pathway and its proportion to the total number of metabolites annotated, while the ordinate is the name of KEGG metabolic pathway.

**Figure 12 plants-11-03390-f012:**
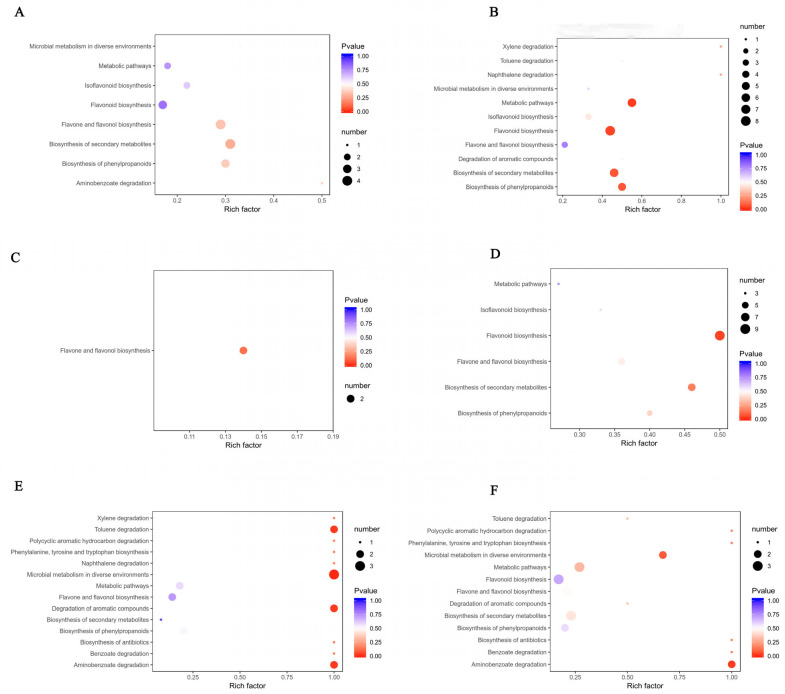
Enrichment results of the flavonoid differential metabolite KEGG pathway in each comparison group. (**A**–**F**) Pathway enrichment results of KEGG of the flavonoid differential metabolites of A1 vs. A6, A1 vs. A6, A1 vs. A2, A2 vs. A3, A3 vs. A4, A4 vs. A5, and A5 vs. A6, respectively. The abscissa is the ratio of the number of differentially expressed metabolites in the corresponding pathway to the total number of metabolites detected and annotated by the pathway, and the ordinate is the name of the KEGG metabolic pathway. The size of the midpoint represents the number of significantly different metabolites enriched in the corresponding pathway.

**Figure 13 plants-11-03390-f013:**
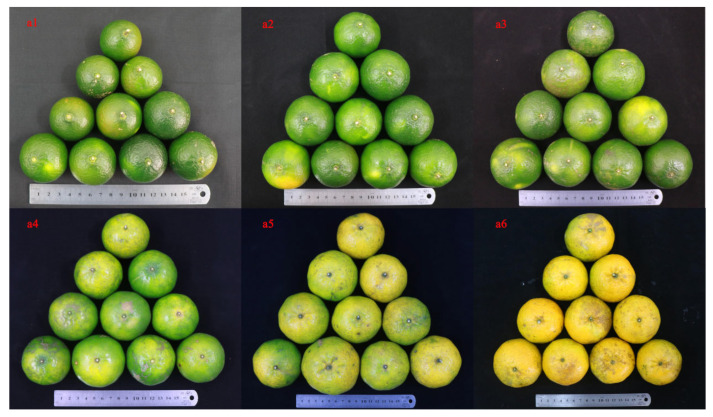
Fresh fruits of *Citrus reticulata* “Chachi” at different harvest times. The plots represent the following: (**a1**) fruit harvested in July is dark green; (**a2**) fruit harvested in August is green; (**a3**) fruit harvested in September is green; (**a4**) fruit harvested in October is yellowish; (**a5**) fruit harvested in November is light yellow; (**a6**) fruit harvested in December is golden yellow.

**Figure 14 plants-11-03390-f014:**
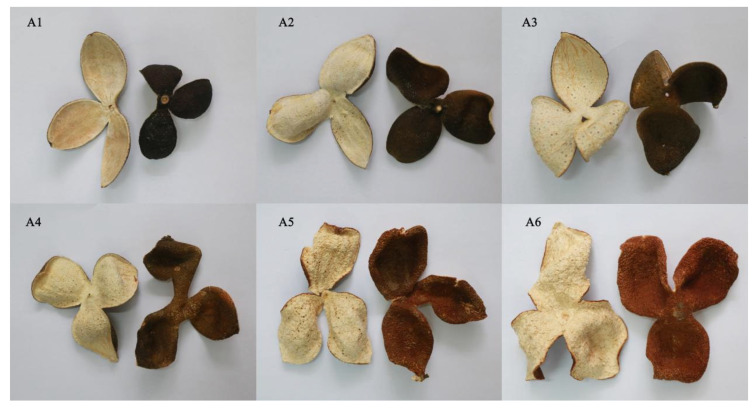
Images of the PCR samples. The plots represent the following: (**A1**) CRPV (Citri Reticulatae Pericarpium Viride) that was harvested and processed in July is cyan black; (**A2**) CRPV that was harvested and processed in August is bluish brown; (**A3**) CRPV that was harvested and processed in September is blue; (**A4**) reddish PCR that was harvested and processed in October is yellowish brown; (**A5**) reddish PCR that was harvested and processed in November is yellowish brown; (**A6**) red PCR that was harvested and processed in December is red brown.

**Table 1 plants-11-03390-t001:** Evaluation of total antioxidant capacity of PCR in different harvest periods. The measurement results of FRAP method and ABTS method are expressed in FeSO_4_ • 7H_2_O and Trolox equivalent, respectively, and the unit is mM. Different letters indicate significant differences between the total antioxidant capacity of PCR of each month. The same letter indicates no significant difference between the total antioxidant capacity of PCR of these months.

Harvest Time	Total Antioxidant Capacity	
	FRAP (mM)	ABTS (mM)
Jul	1.507 ± 0.031 b	0.458 ± 0.084 ab
Aug	2.167 ± 0.025 d	0.639 ± 0.027 c
Sept	1.823 ± 0.153 c	0.631 ± 0.027 c
Oct	1.381 ± 0.125 b	0.601 ± 0.009 bc
Nov	1.104 ± 0.171 a	0.448 ± 0.196 ab
Dec	1.109 ± 0.010 a	0.359 ± 0.022 a

**Table 2 plants-11-03390-t002:** Molecular docking results of positive control drug.

Positive Control Drug	Binding Energy (kcal/mol)			
	3CLpro	RdRp	PLpro	Spike Protein
Lopinavir	−6.20	−10.03	−6.10	−11.40
Ritonavir	−6.0	−8.57	−6.23	−9.37
Ribavirin	−5.87	−7.67	−7.27	−7.43
Chloroquine	−6.07	−6.97	−4.97	−7.13
Arbidol	−5.97	−8.07	−4.90	−6.97

**Table 3 plants-11-03390-t003:** Molecular docking results of flavonoids at 3CLpro.

No.	Flavonoid Name	Binding Energy (kcal/mol)	No.	Flavonoid Name	Binding Energy (kcal/mol)
1	Isoschaftoside	−9.42	14	Ononin	−7.63
2	Vitexin	−8.97	15	Phlorizin	−7.43
3	Narirutin	−8.93	16	Gallocatechin	−7.23
4	Kaempferin	−8.87	17	Nicotiflorin	−7.17
5	Isorhoifolin	−8.77	18	Hesperetin	−7.10
6	Quercitrin	−8.73	19	Luteolin	−6.93
7	Linarin	−8.70	20	Tangeretin	−6.83
8	Neohesperidin	−8.33	21	Nobiletin	−6.80
9	Naringenin 7-O-glucoside	−8.17	22	5,6,7,8,3′,4′ -Hexamethoxyflavanone	−6.70
10	Lonicerin	−7.93	23	Apigenin	−6.67
11	Naringin	−7.83	24	Tectochrysin	−6.53
12	Tiliroside	−7.77	25	Hesperidin	−6.43
13	Cynaroside	−7.70	26	Saponarin	−6.27

**Table 4 plants-11-03390-t004:** Molecular docking results of flavonoids at RdRp.

No.	Flavonoid Name	Binding Energy (kcal/mol)	No.	Flavonoid Name	Binding Energy (kcal/mol)
1	Linarin	−11.97	7	Naringin	−10.83
2	Isorhoifolin	−11.73	8	Saponarin	−10.67
3	Narirutin	−11.63	9	Hesperetin	−10.43
4	Lonicerin	−11.13	10	Neohesperidin	−10.33
5	Isoschaftoside	−11.07	11	Gallocatechin	−10.07
6	Nicotiflorin	−10.93			

**Table 5 plants-11-03390-t005:** Molecular docking results of flavonoids at PLpro.

No.	Flavonoid Name	Binding Energy (kcal/mol)	No.	Flavonoid Name	Binding Energy (kcal/mol)
1	Neohesperidin	−8.00	7	Kaempferin	−7.50
2	Naringin	−7.83	8	Narirutin	−7.50
3	Quercitrin	−7.80	9	Nicotiflorin	−7.37
4	Isorhoifolin	−7.63	10	Gallocatechin	−7.37
5	Linarin	−7.63	11	Cynaroside	−7.30
6	Lonicerin	−7.53			

**Table 6 plants-11-03390-t006:** Molecular docking results of flavonoids at spike protein.

No.	Flavonoid Name	Binding Energy (kcal/mol)	No.	Flavonoid Name	Binding Energy (kcal/mol)
1	Isoschaftoside	−13.33	4	Isorhoifolin	−11.70
2	Lonicerin	−11.90	5	Hesperidin	−11.70
3	Naringin	−11.83			

**Table 7 plants-11-03390-t007:** Information about PCR samples.

No.	Harvest Time	Interval Time (d)
A1	15 Jul 2017	0
A2	15 Aug 2017	30
A3	16 Sept 2017	31
A4	18 Oct 2017	32
A5	23 Nov 2017	35
A6	15 Dec 2017	28

## Data Availability

Data appear within the article and Appendix A.

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
