# Peer review of "Study on Flavonoids and Bioactivity Features of Pericarp of Citrus reticulata “Chachi” at Different Harvest Periods"

_plants, 2022, doi:10.3390/plants11233390_

Round 1

Reviewer 1 Report

Dear authors,

congratulations on a very interesting manuscript. In order to improve it, I recommend the following:

- In materials and methods, many methods have been described the same way as in previous studies (e.g. HPLC conditions and FSI-Q TRAP-MS/MS have even the same sentence order as in "Comparative Analysis of Proanthocyanidin Metabolism and Genes Regulatory Network in Fresh Leaves of Two Different Ecotypes of Tetrastigma hemsleyanum"). Please reorganize the text to avoid plagiarism, regardless that same method was used.

- I recommend writing abbreviations under figures (A1, A2,...) for easier reading, at least a short reminder of what each abbreviation means.

- Line 541, please write capital letter M in word methods, and check the text grammatically.

Reviewer 2 Report

The paper submitted by Liang et al. is focused on the flavonoids content and bioactivity features of pericarp of Citrus reticulata at different harvest periods. The article is well-organised, readable, and clearly prepared. Tables and figures are well-edited and contain relevant information for the readers. However, some corrections and improvements are needed.

Please find my comments below:

The purpose of the paper should be clearly defined in both the introduction and the abstract.

The title of the paper should also refer to the mentioned antiviral properties.

Line 57: “At present…” – new paragraph

Are the results shown in Figure 2 statistically different?

It should be explained why exactly the focus was on antiviral properties. Why not other properties of flavonoids, which are well known and confirmed by a significant amount of research, such as anticancer activity?
